# L-Glutamate Regulates Npy via the mGluR4-Ca^2+^-ERK1/2 Signaling Pathway in Mandarin Fish (*Siniperca chuatsi*)

**DOI:** 10.3390/ijms251810035

**Published:** 2024-09-18

**Authors:** Jiahui Duan, Qiuling Wang, Shan He, Xu-Fang Liang, Liyun Ding

**Affiliations:** 1College of Fisheries, Huazhong Agricultural University, No.1, Shizishan Street, Hongshan District, Wuhan 430070, China; jhd0123@outlook.com (J.D.); qiulingwang_230714@webmail.hzau.edu.cn (Q.W.); heshan@mail.hzau.edu.cn (S.H.); 2Engineering Research Center of Green Development for Conventional Aquatic Biological Industry in the Yangtze River Economic Belt, Ministry of Education, Wuhan 430070, China; 3Poyang Lake Fisheries Research Centre of Jiangxi Province, Jiangxi Fisheries Research Institute, Nanchang 330039, China; dingliyun2008@163.com

**Keywords:** mGluR4, L-glutamate, Npy, mandarin fish (*Siniperca chuatsi*), Ca^2+^ regulation, ERK1/2

## Abstract

Metabotropic glutamate receptor 4 (mGluR4) is widely regarded as an umami receptor activated by L-glutamate to exert essential functions. Numerous studies have shown that umami receptors participate in food intake regulation. However, little is known about mGluR4’s role in mediating food ingestion and its possible molecular mechanism. Mandarin fish, a typical carnivorous fish, is sensitive to umami substances and is a promising vertebrate model organism for studying the umami receptor. In this study, we identified the mGluR4 gene and conducted evolutionary analyses from diverse fish species with different feeding habits. mGluR4 of mandarin fish was cloned and functionally expressed to investigate the effects of L-glutamate on mGluR4. We further explored whether the signal pathway mGluR4-Ca^2+^-ERK1/2 participates in the process in mandarin fish brain cells. The results suggest that L-glutamate could regulate Neuropeptide Y (Npy) via the mGluR4-Ca^2+^-ERK1/2 signaling pathway in mandarin fish. Our findings unveil the role of mGluR4 in feeding decisions and its possible molecular mechanisms in carnivorous fishes.

## 1. Introduction

Sweet, umami, bitter, sour, and salty are classified as five basic taste modalities [1,2], which play vital roles for vertebrates in food intake [3]. Umami is described as a ‘meaty’ perception, and the umami intensity mirrors the nitrogen content, which refers to amino acids [4,5,6]. Researchers have found that metabotropic glutamate receptor 4 (mGluR4) is involved in the umami perception process [7,8]. Two variants of mGluR4, taste-mGluR4 and brain-mGluR4, were the first candidate G protein-coupled taste receptors to be identified in umami taste [8,9,10]. mGluR4 stably responds to L-glutamate, and the affinity of brain-type receptors for glutamate is more than 100 times higher than that of taste-mGluR4 in rats [9,11]. The mGluR4 exists in the brain with the highest relative expression [12,13]. The literature shows that umami receptors regulate food intake [14,15]. However, there is a paucity of information on the involvement of mGluR4 in mediating food ingestion and its possible molecular mechanism.

L-glutamate is a prototypical compound representative of the umami taste, and free L-glutamate in foods elicits umami taste [16]. L-glutamate could be perceived by mGluR4, which has the most sensitive feeling to monosodium glutamate (a typical umami compound) [4]. Researchers found that mGluR4-KO mice had reduced responses to glutamate, and antagonists of mGluR4 could significantly block the responses of L-glutamate in mice’s taste sensory cells [11]. Scientists discovered that L-glutamate rapidly promotes food consumption using a quantitative feeding assay [17]. Studies have shown that fortifying meals with an appropriate amount of monosodium glutamate (MSG) may improve food intake [7]. Neuropeptide Y (NPY) is one of the most effective factors that promotes food intake in mammals and fish [18]. Intracerebroventricular (ICV) injection of Npy promoted feeding in goldfish (*Carassius auratus*) and zebrafish (*Danio rerio*) [19,20]. Some studies have shown that G protein-coupled receptors (GPCRs) have been confirmed to be involved in feeding regulation through the Ca^2+^-ERK1/2 pathway and participate in mediating kinds of hormones in the brain in mammals [21,22]. As a member of the GPCR family, it is likely that mGluR4 also regulates food intake through the Ca^2+^-ERK1/2 signal pathway.

Mandarin fish (*Siniperca chuatsi*) is a unique commercial aquaculture species in China with high nutritional and economic values. Moreover, mandarin fish are extremely carnivorous fish that only accept live prey and refuse dead prey fish or artificial diets their whole lives, and they have a high demand for umami [23]. Therefore, mandarin fish could be a suitable and valuable model organism for investigating the relationship of mGluR4 and Npy with L-glutamate stimulus. To explore the role of mGluR4 in food ingestion, we analyzed mGluR4 genes and conducted evolutionary analyses of different fish species. We also cloned mandarin fish mGluR4 and functionally expressed the gene to prove that mGluR4 can be activated by L-glutamate in mandarin fish. Furthermore, we used the brain cells of mandarin fish to assess the effects of L-glutamate on *npy* via mGluR4 and to explore whether the Ca^2+^-ERK1/2 signaling pathway participates in this process. By integrating the results of the present study, we intend to reveal the role of mGluR4 in feeding decisions in carnivorous fishes and its possible molecular mechanism.

## 2. Results

### 2.1. Synteny, Sequence Alignment, and Phylogenetic Analysis of mGluR4 Genes in Mandarin Fish

We identified the mGluR4 gene in vertebrate genomes and performed a synteny analysis in eight species, including human (*Homo sapiens*), mouse (*Mus musculus*), zebrafish (*Danio rerio*), medaka (*Oryzias liatipes*), grass carp (*Ctenopharyngodon idella*), mandarin fish (*Siniperca chuatsi*), striped sea bass (*Morone saxatilis*), and tilapia (*Oreochromis niloticus*). The results show that human and mouse flanking genes and amino acid sequences were conserved. However, different fish species possessed significantly different gene orders. The adjacent genes of mGluR4 in mandarin fish were more similar to those in striped sea bass, which is also a carnivorous fish (Figure 1 and Figure 2). Mandarin fish mGluR4 shares the highest homology (96%) with the striped sea bass (*Morone saxatilis*) and the grass carp (*Ctenopharyngodon idella*) (86%) and shares 79% of identity among mammalian species (Table 1). Our phylogenetic analysis shows that the tree was divided into two groups: one contains carnivorous fish, and the other contains omnivorous and herbivorous fish (Figure 3).

### 2.2. L-Glutamate Regulates Npy via mGluR4 in Mandarin Fish Brain Cells

The potential role of L-glutamate in regulating Npy in mandarin fish brain cells is shown below. The expression of *npy* increased markedly after 1 h of treatment with 100 mM of L-glutamate (L-Glu) compared with other groups (*p* = 0.001, *df* = 8, and *F* = 29.878). In addition, we found that the mRNA expression of *npy* has an intendency of increasing compared with treatment using 1 mM and 10 mM, but not significantly (*p* > 0.05), which is possibly due to the inter-sample error (Figure 4A). A CCK-8 assay was used to detect the toxic effects of stimulation with different concentrations of L-Glu, and the results suggest that the cell viability in four L-Glu groups (0.1, 1, 10, and 100 mM) was not significantly different compared with the control group (*p* > 0.05) (Figure 4B). Metabotropic glutamate receptor 4 (mGluR4), as a typical umami compound, has the most sensitive feeling to monosodium glutamate. Our results suggest that the relative mRNA expression of mGluR4 was significantly increased when treated with 100 mM of L-Glu in mandarin fish brain cells (*p* = 0.006, *df* = 10, and *F* = 12.483) (Figure 4C). In addition, we used an antagonist of mGluR4 (RS)-alpha-methylserine-O-phosphate (MSOP) to confirm that L-Glu regulates *npy* via mGluR4 in mandarin fish. In our study, the mRNA expression of *npy* stimulated by 100 mM of L-Glu was strikingly reduced when treated with 0.5 mM of MSOP (*p* = 0.017, *df* = 10, and *F* = 8.459) (Figure 4D).

### 2.3. Ca^2+^ Involvement in L-Glutamate Regulating Npy through mGluR4 in Mandarin Fish Brain Cells

We generated intron-less mGluR4 expression constructs in the pcDNA3.1 vector in HEK-293T cells and used the calcium dye fluo-4-am to incubate transfected HEK-293T cells to assay the change in intracellular Ca^2+^ with treatment using 100 mM of L-Glu. The results show that Ca^2+^ release was significantly higher in the mGluR4-pcDNA3.1 vector group than in the pcDNA3.1 vector group after stimulation with 100 mM of L-Glu (*p* = 0.021, *df* = 5, and *F* = 13.500) (Figure 5A,B). Additionally, the mRNA expression of *npy* was detected with the treatment of chelator of intracellular Ca^2+^ BAPTA-acetoxymethyl ester (BAPTA-AM) in mandarin fish brain cells, and the results suggest that the mRNA expression of *npy* was significantly decreased with treatment using 50 µM of BAPTA-AM (*p* = 0.003, *df* = 9, and *F* = 17.359) (Figure 5C).

### 2.4. ERK1/2 Involvement in L-Glutamate Regulating Npy through mGluR4 in Mandarin Fish Brain Cells

To examine whether ERK1/2 plays an important role in the process of Npy regulation via L-Glu, we used an antagonist of mGluR4 MSOP to treat mandarin fish brain cells in the stimulation of L-Glu and detected the phosphorylation level of ERK1/2. In our study, the phosphorylation level of ERK1/2 was significantly increased by L-Glu stimulation (*p* = 0.000, *df* = 9, and *F* = 532.152) while markedly being inhibited by treatment with the MSOP (*p* = 0.000, *df* = 5, and *F* = 332.509) (Figure 6). At the same time, the specific inhibitor of ERK1/2 1,4-Diamino-2,3-dicyano-1,4-bis (o-aminophenyl mercapto) butadiene (U0126) was used to examine the role of ERK1/2. Treatment with the inhibitor of ERK1/2 U0126 for 2 h eliminated the increase in relative mRNA *npy* expression stimulated by L-Glu (Figure 7) (*p* = 0.025, *df* = 11, and *F* = 6.959).

## 3. Discussion

The sense of taste is one of a chemosensory system that perceives nutritional and harmful substances in food, and the taste of umami could help vertebrates recognize L-amino acids [24]. The umami taste is elicited by L-glutamate and other amino acids which are natural constituents of many protein-rich foods. Metabotropic glutamate receptor 4 (mGluR4) has been demonstrated to be important in sensing umami taste, especially L-glutamate in mammals [11]. Zhao et al. (2010) illustrated that the absence of the umami taste *Tas1r1* results from giant pandas’ decreased reliance on meat, which, in turn, reinforced their herbivorous lifestyle [25]. Wolsan and Sato (2022) propose that feeding specializations trigger a time-dependent process of taste receptor loss through the deprivation of using gustatory receptors, and this process may be decelerated/stopped because of the extra gustatory functions of the receptors’ protein(s) through observing the evolution of the sweet and umami receptors in Carnivora (dogs, cats, and kin) [26]. Scholars have classified fish feeding habits as herbivorous, carnivorous, and omnivorous. We hypothesized a correlation between mGluR4 and food habits in fish, which affect food intake. Thus, we performed synteny, sequence alignment, and a phylogenetic analysis of mGluR4 genes from different feeding habits of fish species. The results show that the carnivorous fish had the highest homology. Based on this, we further explored the role of mGluR4 in feeding regulation in extreme carnivorous mandarin fish at the cellular level.

Several studies in mammals have reported that umami receptors regulate food intake stimulated by umami compounds in the intestine [14]. Metabotropic glutamate receptor 4 (*mGluR4*), expressed in a region-specific way, acts as a detector for nutrients in the stomach and gastric ghrelin cells [27]. Exogenous nutritional stimuli seem to be transformed into neuronal inputs through nutrient-sensing receptors, and then the nutritional status is transformed into particular appetites between the gut and brain [28]. The orexigenic Neuropeptide Y (*Npy*) expression levels were promoted slightly but significantly with 7 days’ monosodium glutamate treatment in adult male rat pituitary [29]. Supplementing L-glutamate in the diet could significantly increase the feeding rate in Atlantic salmon (*Salmo salar*) and Jian carp (*Cyprinus carpio*) [30,31]. Similar to the results reported in fish before, our study found that L-glutamate could significantly promote the mRNA expression of *npy*, suggesting that *npy* could be regulated by L-glutamate in mandarin fish. However, ICV injection of L-glutamate significantly suppressed feeding in broiler chicks and promoted the mRNA levels of a hormone that decreases appetite [32]. The different results obtained in previous studies might be attributed to the different species and routes of administration.

Yasumatsu et al. (2015) suggested that *mGluR4* knockout mice showed decreased responses to L-glutamate in the glossopharyngeal and chorda tympani nerves [11]. L-glutamate supplements in weaning piglets promoted the *mGluR4* expression of jejunal mucosa [33]. However, whether the umami receptor mGluR4 regulates Npy and controls food intake in mandarin fish has not been well characterized. In the present study, we explored the molecular mechanism of L-glutamate and how it mediates feeding via *mGluR4* in the brain cells of mandarin fish. Our research shows that the *mGluR4* expression was significantly increased by 100 mM of L-Glu, suggesting that *mGluR4* could respond to L-Glu in mandarin fish. It was found that the use of MSOP, a selective antagonist of mGluR4, can significantly and reversibly suppress responses evoked by L-amino acids [8]. Research scholars have reported that mGlur4 decreased the probability of glutamate release from mossy fiber (MF) terminals synapsing onto Stratum lacunosum-moleculare interneurons (L-Mi), while the application of mGluR4 antagonist MSOP (100 μM) revealed a relief of inhibition of transmitter release [34]. Thus, we used MSOP to confirm the potential role of mGluR4 in the regulation of Npy via L-glutamate in the brain cells of mandarin fish. Our data indicate that MSOP could eliminate the increased expression of *npy* stimulated by L-Glu. The mRNA expression of *npy* decreased to about 32–41% of the control level after treatment with the agonist of mGluR4 (1S,3R,4S)-1-aminocyclopentane-1,3,4-tricarboxylic acid (ACPT-1) in rats [35]. In combination with the described results, we propose that mGluR4 plays an essential role in the regulation of NPY via L-glutamate in mandarin fish.

As an important part of the signal network, the Ca^2+^-ERK1/2 signaling pathway has been confirmed to be involved in feeding regulation in mammals [22]. The trigger of signal transduction pathways of different tastes such as sweet, umami, and salty ultimately promote Ca^2+^ release, indicating the significance of Ca^2+^ in the transduction of taste receptors [36,37,38]. Neuromedin U receptor 2 (NMU2), a G protein-coupled receptor, has been discovered to mediate feeding through Ca^2+^ in human embryonic kidney 293 cells [21]. Bombyx neuropeptide GPCR A15 regulates feed intake by stimulating Ca^2+^ mobilization and ERK1/2 phosphorylation in bombyx [39]. The calcium concentration was enhanced by stimulating L-glutamate in hippocampal neurons [35]. To explore whether mGluR4 also transmits signals via calcium ions in the presence of amino acids, we generated intron-less mGluR4 expression constructs in HEK-293T cells and utilized the calcium dye fluo-4-am to incubate the cells to detect the release of intracellular Ca^2+^ with stimulation using 100 mM of L-Glu. We found that the release of Ca^2+^ was significantly higher in the mGluR4 expression constructs group than in the control group with stimulation using 100 mM of L-Glu. The results strongly suggest that the response to L-Glu occurred via mGluR4. The hippocampal neurons of rats secreted NPY with the stimulation of glutamate in a Ca^2+^-dependent way [40]. Therefore, we used the BAPTA-AM inhibitor to verify the Ca^2+^ involved in the process of Npy regulation via L-Glu through mGluR4 in mandarin fish. An amount of 50 µM of BAPTA-AM eliminated the increased expression of *npy* activated by L-Glu, indicating that Ca^2+^ was involved in the process described above.

ERK1/2, belonging to MAPK, is crucial in the signaling pathway that transmits nutritional signals from kinds of extracellular factors to mediate proliferation differentiation [41]. After receiving external stimulation such as glutamate, Ca^2+^ signaling could immediately activate the phosphorylation level of ERK1/2 [42,43]. A variety of intracellular signaling pathways containing G protein-coupled receptors require the participation of ERK1/2 [44]. mGluR4 activates the phosphorylation of ERK1/2 in the embryonic neural progenitor cells of rats [45]. In our study, consistent with mammals, the phosphorylation levels of ERK1/2 could be activated by L-Glu, and 0.5 mM of MSOP significantly eliminated the increased phosphorylation levels of ERK1/2. To further investigate the potential role of ERK1/2 in the process of Npy regulation via L-glutamate, we used the inhibitor of ERK1/2 U0126. The inhibitor eliminated the increased mRNA expression of *npy* stimulated by L-Glu, suggesting that L-Glu might regulate *npy* expression through ERK1/2. ERK1/2 signaling is also demanded to stimulate the *Npy* gene and contributes to a fasting response [46,47]. Similar results were found in rats in which the secretion of NPY was partially inhibited by the inhibitor of the phosphorylation of ERK in cells separated from the cortex [48]. The capability of CCK to inhibit feeding was blocked by cutting off the ERK1/2 signaling cascade in rats [49].

## 4. Materials and Methods

### 4.1. mGluR4 Sequence Analysis

The sequence of the mGluR4 gene of humans (*Homo sapiens*), mice (*Mus musculus*), zebrafish (*Danio rerio*), Japanese medaka (*Oryzias latipes*), Mexican tetra (*Astyanax mexicanus*), striped sea bass (*Morone saxatilis*), Nile tilapia (*Oreochromis niloticus*), coelacanth (*Latimeria chalumnae*), channel catfish (*Ictalurus punctatus*), Torafugu (*Takifugu rubripes*), Chinook salmon (*Oncorhynchus tshawytscha*), barramundi perch (*Lates calcarifer*), grass carp (*Ctenopharyngodon idella*), mandarin fish (*Siniperca chuatsi*), clown anemonefish (*Amphiprion ocellaris*), common carp (*Cyprinus carpio*), Wuchang bream (*Megalobrama amblycephala*), kelp grouper (*Epinephelus moara*), and Burton’s mouthbrooder (*Haplochromis burtoni*) were obtained from GenBank “https://www.ncbi.nlm.nih.gov/genbank/ (accessed on 4 November 1988)”. Accession numbers of sequences are listed in Appendix A. Synteny and multiple amino acid sequence alignment analysis were performed by screening mGluR4 flanking genes and Clustal X in zebrafish, medaka, Nile tilapia, grass carp, striped sea bass, mice, and humans. The mGluR4 amino acid sequences were obtained from the National Center for Biotechnology Information (NCBI) genome browser. Phylogenetic tree analyses were conducted using MEGA7 by using neighbor-joining tree evaluated with 1000 bootstrap replicates.

### 4.2. Fish and Sample Collection

The mandarin fish were bred at the Chinese Perch Research Center of Huazhong Agricultural University, Wuhan, China. All animal experiments were permitted by the Ethics Committee of Huazhong Agricultural University. The adult mandarin fish brain was collected after treatment with MS-222 for primary culture (see below).

### 4.3. Mandarin Fish Brain Cell Culture and Treatment

The mandarin fish brain was isolated after treating with MS-222 and soaked in the L15 cell-treated buffer containing antibiotics and antimycotics (Invitrogen, Waltham, MA, USA) at a final concentration of 1000 IU/mL penicillin, 1000 μg/mL streptomycin, 250 μg/mL amphotericin B, and 50 mg/mL in a dish for 1 h. Then, the brain was minced thoroughly and transferred into a new 15 mL eppendorf tube supplemented with type I collagenase to digest into brain cells for 2.5 h. The mandarin fish brain cells were centrifuged at 1000 r/min for 5 min at room temperature and then resuspended and cultured by L15 cell-treated buffer (Leibovitz, Beijing, China) supplemented with 20% FBS (Gibco, Grand Island, NY, USA) and 1% bioantibiotics (penicillin-streptomycin) (Gibco, Grand Island, NY, USA) at 28 °C without CO_2_ for 4 days. Then, subculturing of cells was carried out every two days. For the experiment, the cells were seeded on 6 cm dishes at a density of 1.0 × 10^5^ in growth medium. Two days later, the cells were cleaned twice with a phosphate-buffered solution (DBPS) and hungered with L15 cell-treated buffer without serum for 24 h when the coverage was 80%. The cell culture approach follows standard practices in studies involving mixed brain cell cultures; the term ‘mandarin fish brain cells’ refers to a heterogeneous mixture of different cell types found in the brain. Cell cultures include various types of neurons and glial cells without distinguishing between these cell types at the individual level. The cells were treated with L15 as the control group or with different concentrations of L-glutamate (0, 0.1, 1, 10, and 100 mM) for 1 h as the L-Glu group. The brain cells were pretreated for 1 h with the antagonist of mGluR4 MSOP, 15 min with the Ca^2+^ chelator BAPTA-AM, and 2 h with ERK1/2 inhibitors U0126 separately before the 60 min L-glutamate (L-Glu) stimulus period.

### 4.4. Cell Counting Kit-8 (CCK-8)

The CCK-8 assay was used to assess cell activity and survival. The mandarin fish brain cells were placed into 96-well plates at a density of 5 × 10^5^ per mL. A volume of 100 μL of cell suspension was added to each well in six replicates in each group. The cells were allowed to stand for 24 h under a standard culture environment, and then different concentrations of L-glutamate media were added to each group continually for 12 h. After the indicated treatment, 10 μL of CCK-8 solution (Abbkine, Wuhan, China) was added to each well, and the plates were incubated at 28 °C for an extra 4 h. Cell activity was assessed by detecting absorbance value at 450 nm with the application of a multifunctional microplate reader (BioTek Instruments, Inc., Winooski, VT, USA).

### 4.5. Cloning mGluR4 Gene Sequences of Mandarin Fish

Total RNA was extracted from the brains of mandarin fish using Trizol Reagent (TaKaRa, Tokyo, Japan). cDNA was obtained using HiScript II Q RT SuperMix reverse transcriptase (Vazyme, Nanjing, China). All deduced exons of mGluR4 were amplified by overlapping PCR to obtain the full-length mGluR4. The overlapping PCR primers amplified the full-length sequence of mandarin fish mGluR4 (Table 2). The target fragment was recovered and purified using Omega EZNA Gel Extraction Kit (Omga, Norwalk, CT, USA). The entire coding sequences were ligated to the restriction sites of the pcDNA3.1 cloning vector (TransGen Biotech, Beijing, China), and then the plasmid was transferred into DH5αcompetent cells (TransGen Biotech, Beijing, China). The positive strains were preserved and sequenced by Sangon Biotech Company (Shanghai, China).

### 4.6. mGluR4 Functional Expression

For the mandarin fish mGluR4 assay, intronless mGluR4 expression construct (mGluR4-pcDNA3.1 Vector) was generated and transfected into HEK-293T cells using Lipofectamine 2000 (Invitrogen, Carlsbad, CA, USA). HEK-293T cells were grown and maintained at 37 °C at 5% CO_2_ in DMEM supplemented with 10% fetal bovine serum (FBS, Gibco, Grand Island, NY, USA). The HEK-293T cells were seeded onto a 20 mm Confocal Dish with a density of 1.0 × 10^6^ (Vazyme, Nanjing, China) and incubated after 16 h with 80% confluence at 37 °C for treatment.

### 4.7. Calcium Assays

HEK-293T cells, transfected with mGluR4-pcDNA3.1 vector and pcDNA3.1 vector individually, were washed three times with HBSS and loaded with the calcium dye fluo-4-am (YESEN), at an amount of 4 µM in 1 mL HBSS, for 30 min at 37 °C. Then, the culture medium was replaced by 1 mL of HBSS for about 30 min at room temperature, and the cells loaded with a calcium fluorescence probe were subjected to Ca^2+^ imaging. The fluorescence images were measured at 480 nm excitation and 530 nm emission. The cultured cells were stimulated with 100 mM L-Glu, and calcium mobilization was monitored every 2.5 s for an additional 120 sec. Data were compiled from three biological replicates and analyzed with Image J software (×64) 1.6.0.

### 4.8. RT-PCR

Total RNAs of brain cells were extracted using Trizol Reagent (TaKaRa, Tokyo, Japan) as described in the instruction manual and then frozen at −80 °C. Then, the BioTek Synergy™2 Multi-detection Microplate Reader (BioTek Instruments, Winooski, VT, USA) and agarose gel electrophoresis were used to quantify the purity of RNA. cDNA was obtained using HiScript II Q RT SuperMix reverse transcriptase (Vazyme, Nanjing, China). Gene-specific primers of mandarin fish were designed with Primer 5.0 software and synthesized by Sangon (Shanghai, China). The amplification information of the primers is shown in Table 3. The ribosomal protein L13a (*rpl13a*) gene was selected as an internal reference gene because of its stability and rapid amplification. Real-time qPCR was detected using a quantitative thermal cycler (BIO-RAD, Hercules, CA, USA), and 10 μL AceQ^®^ qPCR SYBR^®^ Green Master Mix (Vazyme, China), 8.2 μL double-distilled water, 1 μL template (cDNA), and 0.4 μL forward and reverse primers, respectively, constituted the reaction volume (20 μL). The PCR parameters were 95 °C for 5 min, followed by 40 cycles at 95 °C for 10 s, and then Tm for 30 s and a melt curve step (0.5 °C/s, gradually reducing to Tm, with acquisition data obtained every 6 s). Gene expression levels were quantified relative to the expression of *rpl13a* using the optimized comparative Ct (2^−ΔΔCt^) value method. The fold-change values are the averages of six biological replicates in each group. Data are presented as mean ± SEM (n = 6).

### 4.9. Western Blot

Protein was extracted from the brain cells of mandarin fish, and protein concentrations were determined using a bicinchoninic acid (BCA) protein assay (Biosharp, Beijing, China). The 30 µL protein was separated by 10% SDS-PAGE gel and then transferred onto polyvinylidene difluoride (PVDF) membranes (Millipore, Burlington, MA, USA). Phospho-ERK1/2 and β-tubulin were detected by immunoblotting with rabbit anti-pERK1/2 antibody (1:2000, Cell Signaling Technology, Beverly, MA, USA, #4370) and mouse anti-β-tubulin antibody (1:10,000, Cell Signaling Technology, Beverly, MA, USA, #2146), respectively. Blots were probed by goat anti-rabbit (1:10,000, Cell Signaling Technology, Beverly, MA, USA, #5151) and anti-mouse (1:10,000, Cell Signaling Technology, Beverly, MA, USA, #5470) secondary antibodies with IR-Dye 680 or 800cw label. The results were scanned using a LiCor Odyssey scanner and then analyzed using ImageJ software (×64) 1.6.0. The phosphorylation levels of ERK1/2 were normalized according to the loading of proteins with the data expressed as a ratio of p-ERK1/2 to β-tubulin. Data were compiled from six biological replicates and are presented as mean ± S.E.M. (n = 6).

### 4.10. Statistical Analysis

All experimental data are presented as mean ± S.E.M. The SPSS 19.0 software (Chicago, IL, USA) was applied for statistical analysis. The data were assessed using one-way ANOVA after Levene’s test for homogeneity of variance, and differences in between group means were compared using Duncan’s post hoc multiple comparisons. *p* < 0.05 was defined as statistically significant.

## 5. Conclusions

In summary, we discovered that L-glutamate can regulate *npy* expression through the G protein-coupled receptor mGluR4. Phylogenetic and multiple amino acid alignment analyses of mGluR4 offer the insight that there might be a correlation between mGluR4 and food habits in fish. Our findings indicate that L-glutamate can regulate Npy via the mGluR4-Ca^2+^-ERK1/2 signaling pathway. The present research reveals the role of mGluR4 in feeding decisions and its molecular mechanism.

## Figures and Tables

**Figure 1 ijms-25-10035-f001:**
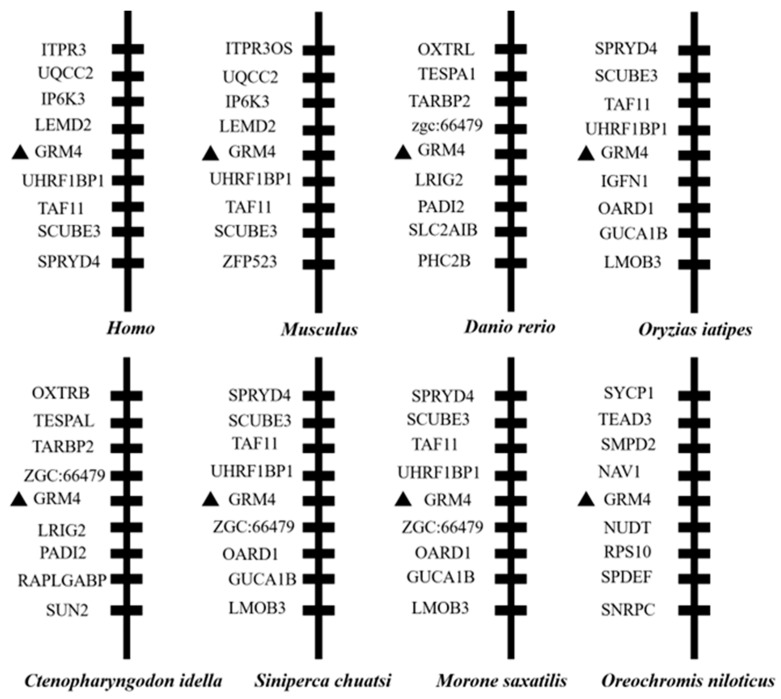
Synteny analysis of mGluR4 gene. Synteny analysis was performed by searching gene flanking mGluR4 in genomes of humans, mice, zebrafish, medaka, grass carp, mandarin fish, striped sea bass, and Nile tilapia using map viewer of National Center for Biotechnology Information (NCBI) genome browser. Mandarin fish mGluR4 is marked with a black triangle.

**Figure 2 ijms-25-10035-f002:**
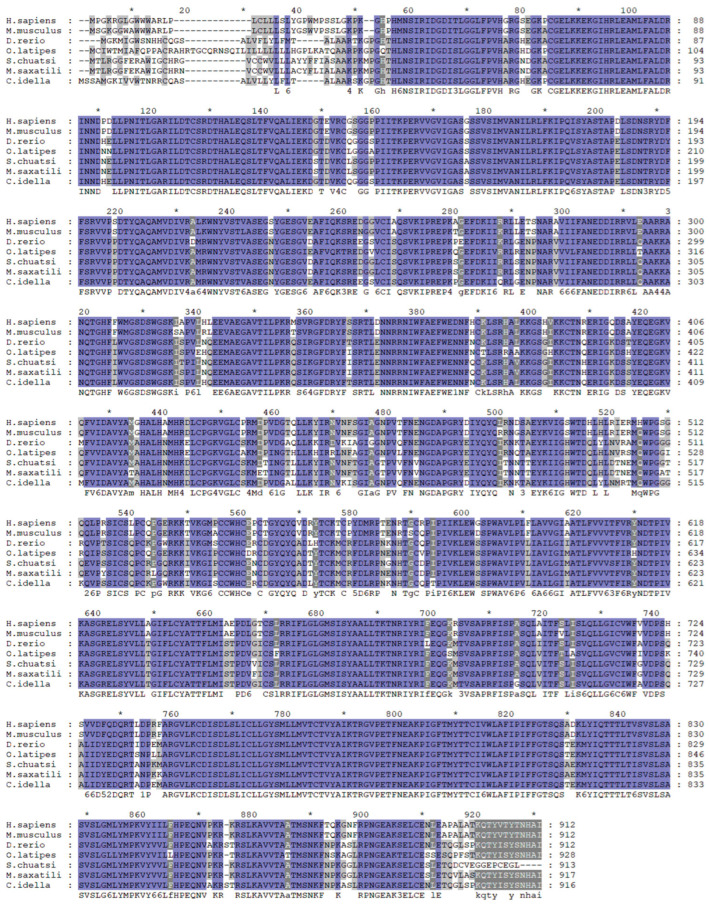
Alignment analysis of mGluR4 genes in humans (*H. sapiens*), mice (*M. musculus*), zebrafish (*D. rerio*), medaka (*O. liatipes*), mandarin fish (*S. chuatsi*), striped sea bass (*M. saxatilis*), and grass carp (*C. idella*). Purple highlights with black letters represent fully conserved residues. Gray highlights with black letters represent conservation between groups with strongly similar properties. * represents the average of the number of amino acids shown on both sides.

**Figure 3 ijms-25-10035-f003:**
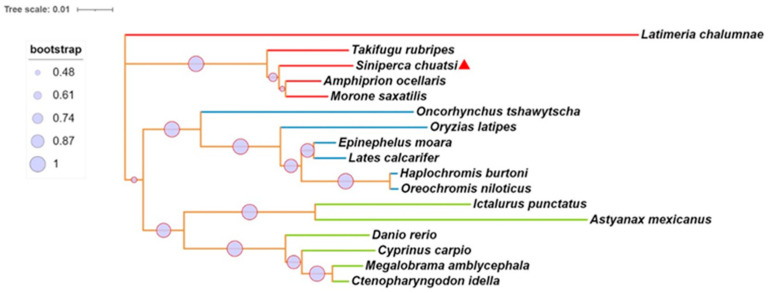
Phylogenetic and molecular evolutionary analyses of mGluR4 genes were conducted according to the amino acid sequences of the pancreatic polypeptide family by MEGA7. Lines with the same color (red, blue, or green) belong to one branch. Mandarin fish mGluR4 is marked with a red triangle.

**Figure 4 ijms-25-10035-f004:**
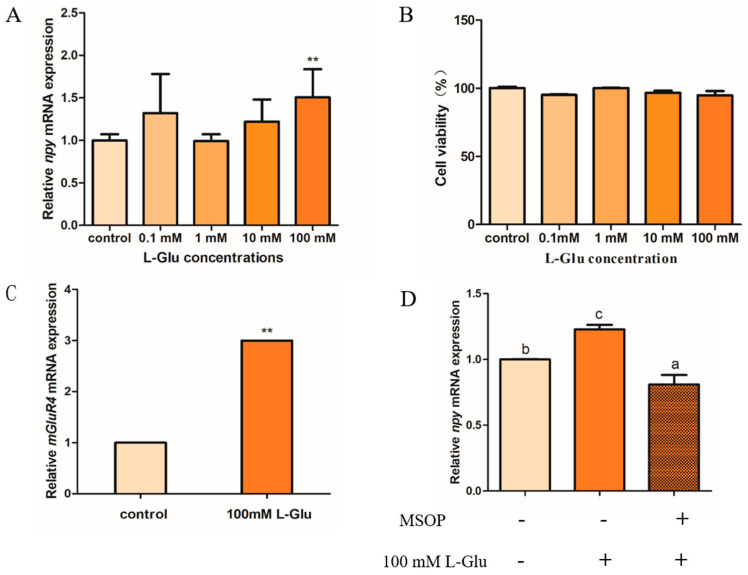
L-glutamate regulates Npy via mGluR4 in mandarin fish brain cells. (**A**) The effect of different concentrations of L-Glu (0.1, 1, 10, and 100 mM) stimulus for 1 h on the relative mRNA expression of *npy* in mandarin fish brain cells (**B**) The cell viability of different concentrations of L-Glu (0.1, 1, 10, and 100 mM) stimulus in mandarin fish brain cells. (**C**) The effect of 100 mM of L-Glu stimulus for 1 h on the relative mRNA expression of *mglur4* in mandarin fish brain cells. (**D**) The effect of MSOP on the relative mRNA expression of *npy*. An amount of 0.5 mM of MSOP was added to mandarin fish brain cells in a complete growth medium 15 min prior to the treatment with 100 mM of L-Glu for 1 h. The effect of 100 mM of L-Glu alone for 1 h or 50 mM of MSOP pretreatment for 1 h on the mRNA expression of *npy*. RNA was isolated for a quantitative polymerase chain reaction (Q-PCR) analysis. All values are presented as the mean ± SEM (n = 6). ** indicate significant differences (*p* < 0.01) between the experiment group treated with L-Glu and the control group, respectively. Vertical bars with different lowercase letters represent a significant difference between each group.

**Figure 5 ijms-25-10035-f005:**
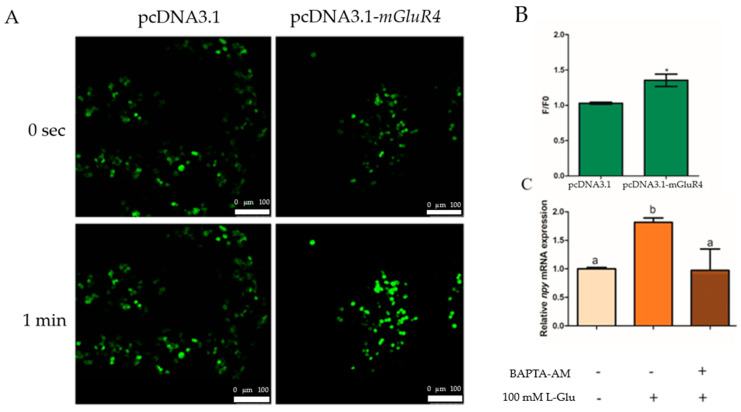
Ca^2+^ involvement in the regulation of Npy via L-glutamate through mGluR4 in mandarin fish brain cells. (**A**) Fluo-4AM-loaded HEK293T cells transfected with pcDNA3.1 or pcDNA3.1-mGluR4 were stimulated by 100 mM of L-Glu. Fluorescent changes are represented in fluo-4 fluorescence images of the cells obtained at 0 s (top panel) and 1 min (bottom panel) after the stimulation, respectively. (**B**) Fluo-4 fluorescence was quantified as the ratio of the fluorescence intensity value of a single cell to its initial intensity value at 0 s and 1 min (F/F0). (**C**) The effect of BAPTA-AM on the relative mRNA expression of *npy*. An amount of 50 μM of BAPTA-AM was added to brain cells in the complete growth medium 15 min prior to the treatment with 50 mM of L-Lys for 1 h. RNA was isolated for a quantitative polymerase chain reaction (Q-PCR) analysis. All values are presented as the mean ± SEM from three independent experiments. * represents the significant difference between the experiment group treated with L-Glu and the control group. Vertical bars with different lowercase letters represent a significant difference between each group (*p* < 0.05).

**Figure 6 ijms-25-10035-f006:**
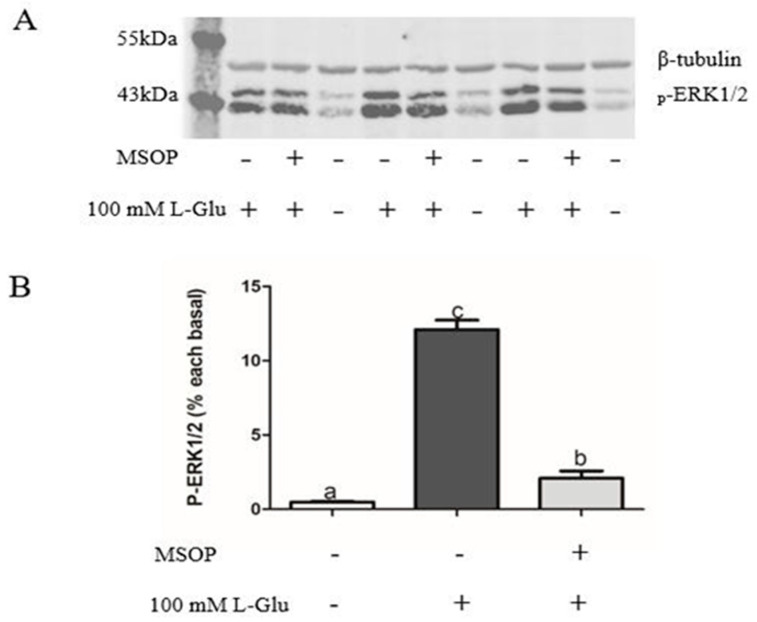
The effect of MSOP on the phosphorylation of ERK1/2 in mandarin fish. (**A**) Pretreatment with 0.5 mM of MSOP in mandarin fish brain cells for 1 h before stimulation with 100 mM of L-Glu. The cells were collected to analyze the phosphorylation levels of ERK1/2 by Western blotting. (**B**) The data of the Western blot was quantified using Image J software (×64) 1.6.0. All values are presented as the mean ± SEM (n = 6). Vertical bars with different lowercase letters represent a significant difference between each group (*p* < 0.05).

**Figure 7 ijms-25-10035-f007:**
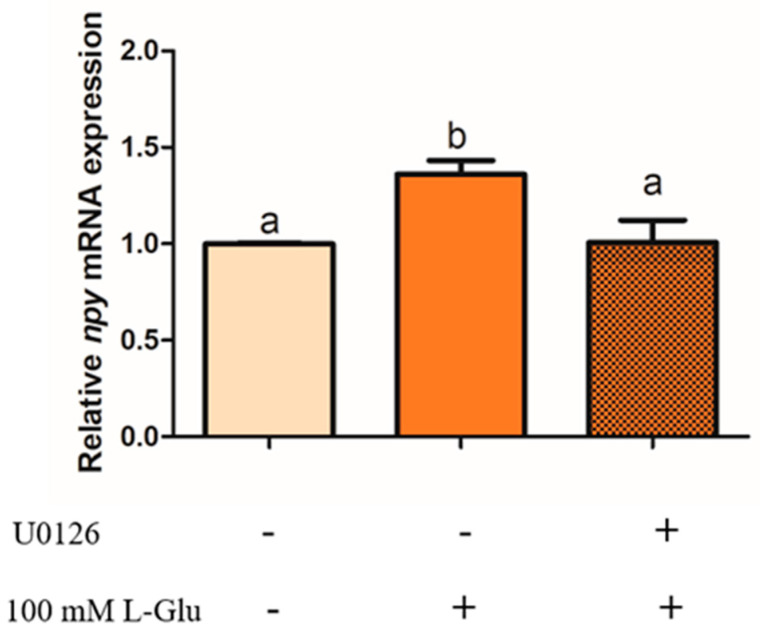
The effect of U0126 on the relative mRNA expression of *npy* in mandarin fish. Pretreatment with 10 μM of U0126 in mandarin fish brain cells for 2 h before 1 h stimulation with 100 mM of L-Glu. The cells were collected for a quantitative polymerase chain reaction (Q-PCR) analysis. All values are presented as the mean ± SEM (n = 6). Vertical bars with different lowercase letters represent a significant difference between each group (*p* < 0.05).

**Table 1 ijms-25-10035-t001:** Amino acid percent homology of mGluR4 between mandarin fish and other vertebrates.

Species	*Siniperca chuatsi* (%)
*Homo sapiens*	79
*Mus musculus*	79
*Danio rerio*	85
*Oryzias latipes*	84
*Morone saxatilis*	96
*Ctenopharyngodon idella*	86

**Table 2 ijms-25-10035-t002:** The nucleotide sequences of the primers for overlapping PCR.

Gene Name Primer	Sequence of Primer (5′ to 3′)	Tm (°C)
*mGluR4*-pcDNA3.1	F TAGTCCAGTGTGGTGGAATTC ATGACTCTACGAGGAGGTTTTGA R GGTTTAAACGGGCCCTCTAGA TTAGATGGCATGGTTACTGTAGC	58

**Table 3 ijms-25-10035-t003:** Primer sequences for quantitative real-time PCR.

Gene Name (%)	Sequence of Primer (5′ to 3′)	Tm (°C)	Amplification Efficiency
*rpl13a*	F TATCCCCCCACCCTATGACA R ACGCCCAAGGAGAGCGAACT	58.0	104.8
*mGluR4*	F GAGTGGTGGGAGTGATTGGTG R GGGAGAAGAAGTCATAGCGGG	58.0	96.4
*npy*	F GTTGAAGGAAAGCACAGACA R GCTCATAGAGGTAAAAGGGG	60.0	92.3

## Data Availability

All data are available from the corresponding author by request.

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
