# Peer review of "L-Glutamate Regulates Npy via the mGluR4-Ca2+-ERK1/2 Signaling Pathway in Mandarin Fish (Siniperca chuatsi)"

_ijms, 2024, doi:10.3390/ijms251810035_

Round 1

Reviewer 1 Report (New Reviewer)

Comments and Suggestions for Authors

Duan et al. conduct an investigation into the role of L-glutamate and mGluR4 in regulating Npy expression through the Ca2+-ERK1/2 signaling pathway in mandarin fish. While the study offers valuable insights into the molecular mechanisms of feeding behavior in fish, the manuscript faces several challenges concerning scientific rigor, clarity, and depth of analysis. The logical progression of the study is occasionally unclear, and the connection between the experiments and the central hypothesis could be more robust. Furthermore, some methodological details are insufficiently described, and the conclusions may be somewhat overstated relative to the data presented.

Specific Comments:

Introduction

Lines28:The phrase "As is known to all" is too informal for a scientific paper.

Lines41-55:The introduction lacks a logical flow, jumping between topics without clear transitions.

Lines67-68:The introduction could benefit from a clearer statement of the hypothesis and research objectives. While the study's goals are mentioned, they are somewhat buried within the text.

Results

Lines69-82:Homology analysis was carried out for part 2.1 lacking Marine fishes.

Lines101-114:In Figure 4A, the higher response of 0.1 mM L-Glu compared to 1 mM and 10 mM L-Glu is not explained or discussed.

Lines132-134:The experimental description does not mention how many times the experiments were repeated, affecting the reliability of the results.

M&M

Lines208:Different concentrations of L-glutamate (0.1, 1, 10, 100 mM) were selected in the experiment, but it was not fully explained why these concentration ranges were selected, whether they were based on previous experiments or literature data.

Lines200-204:In section 3.2, the culture conditions for Mandarin fish brain cells lack detail, particularly regarding CO2 levels and culture time.

Lines223-233:The extraction of total RNA from brain cells of Mandarin fish was mentioned in section 3.4, but the determination method of RNA purity and concentration was not mentioned, which is crucial for subsequent experiments.

Lines266:The statistical methods were described in detail, such as "The data were analyzed by one-way analysis of variance (ANOVA), the differences between groups were tested by LSD method after the fact, and P < 0.05 was considered statistically significant."

Discussion

The discussion on the mGluR4 antagonist MSOP is brief and lacks an in-depth analysis of its biological significance. The discussion does not adequately emphasize the novel contributions of the study or what makes it unique compared to existing research. Some citations in the discussion section are either not strongly related to the study or appear repetitive, such as those referencing the loss of umami receptors in giant pandas, which is not directly relevant to Mandarin fish research. Lastly, the discussion on the MSOP experiment could benefit from a deeper analysis and comparison with other studies to strengthen the argument.

Comments on the Quality of English Language

Moderate editing of English language required.

Author Response

Reviewer 2 Report (New Reviewer)

Comments and Suggestions for Authors

The manuscript by Duan et al. aimed to investigate the role of metabotropic glutamate receptor 4, known as an umami receptor, in regulating food intake in Mandarin fish. The authors identified and cloned the mGluR4 gene and conducted evolutionary analyses across fish species with different feeding habits, and explored the functional effects of L glutamate on mGluR4. Furthermore, they specifically examined whether the mGluR4-Ca2+-ERK1/2 signaling pathway regulates neuropeptide Y (Npy), a key molecule in appetite control. This study reveals that L glutamate can influence feeding decisions in carnivorous fish through signaling pathways, shedding light on the molecular mechanisms behind food ingestion in these species. Generally speaking, this study is interesting; however, it needs minor revision to be reconsidered for publication in IJMS. Below are my comments for the authors to address.

General comments:

1. All abbreviations, including “MSOP” should be defined at their first appearance in the manuscript

2. Authors should modify the materials and method section. There is a need to add a “Fish and sample collection” sub-section. Where did the authors obtain experimental fish? How many were purchased or farmed? Were females or males?

3. Line 282, I would like to know why the authors displayed the data as mean ± standard error (mean ± SEM) rather than mean ± SD.

4. Did the authors submit the cloned gene sequence to the NCBI? The manuscript does not record the NCBI accession number.

5. More information regarding how primers were designed must be added to the MS

6. There are many grammar and format mistakes, e.g., spacing, were found. Please check carefully. I listed some of them below.

Specific comments:

1.     Line 5-11: Replace (;) with (.) at the end of all affiliations.

2.     Line 16: Delete “as”, Replace “to study” with “for studying”

3.     Line 20: Replace “impacts” with “effects”

4.     Line 23 should read as “Our findings unveiled the role of mGluR4 in feeding decisions and its possible molecular mechanisms in carnivorous fishes”.

5.     Line 25: Remove space between (Siniperca chuatsi) and (;).

6.     Line 31: Delete “food”

7.     Line 36: Replace “rat” with “rats”

8.     Line 38 should read as “However, there is a paucity of information on the involvement of mGluR4 in mediating food ingestion and its possible molecular mechanism”

9.     Line 41 should read as “L-glutamate is a prototypical compound representative of the umami taste, and free L-glutamate in foods elicits umami taste”

10.  Line 44 should read as “Researchers found……”  instead of “The Researchers……”

11.  Line 55: Replace “probable” with “likely”

12.  Line 61: Maintain the writing style of “mGluR4” to the whole manuscript.

13.  Line 62: Replace “from different fish species” with “of different fish species”

14.  Line 64 should read as” Furthermore, the brain cells of mandarin fish were used to assess the effects of L-glutamate on npy via mGluR4 and to explore whether the Ca2 +-ERK1/2 signaling pathway participates in this process”.

15.  Line 71 should read as”…………and performed synteny analysis…………” instead of “…….had a synteny analysis………..”  

16.  Line 71-82: Scientific names like Danio rerio, Oryzias latipes, Siniperca chuatsi, etc should be italicized, and authors should cross-check all scientific names and write them correctly. also “Oryzias iatipes” should read as “Oryzias latipes”.

17.  Line 73, 189: “grass” should be replaced with “grass carp”.

18.  Replace “Figure 2, Figure 3” with “Figure 2, Figure 3”.

19.  Line 77 should read as “…… to those in striped sea bass………”

20.  Line 79: Delete “that of”, also replace “the grass” with “grass carp”

21.  Line 89: Figure 2, write the full name of the species on the left side of multiple sequence alignment. What do numbers on the right side represent?

22.  Figure 3: The Figure’s caption should explain all details in a particular figure; for example, what does red triangle, red lines, blue lines, and green lines indicate?

23.  Table 1 can be placed as supplementary data

24.  Table 3: What do you mean by “Primer Sequence of primer (5’ to3’)? and “amplification efficiency (%) should be written in the same table column

25.  Line 99: Replace “Chinese perch” with “mandarin fish”.

26.  Line 102: Replace “was” with “is”.

27.  Line 102” For “L-Glu”, did you mean “L-Glutamate”?

28.  Line 103: Add “of” before the word “treatment”

29.  Line 109: replace “was largely” with “ was significantly”

30.  Line 125: Replace “indicated” with “indicates”.

31.  Line 126: Replace “represented” with “represents”.

32.  Line 130: Replace “We have” known…….with “It is” known……

33.  Line 134 should read as “The results showed that Ca2 + release was significantly higher in the mGluR4-pcDNA3.1 vector group than in the pcDNA3.1 vector group after stimulation with 100 mM L-Glu.”…..

34.  Line 158: Replace.... “with the L-Glu stimulation”…..with… “by L-Glu stimulation”……

35.  Line 161: Maintain the writing style of the NPY gene, it is either “npy” or “Npy”, this problem should be addressed throughout the whole manuscript.

36.  MSOP should be defined at the first appearance in the manuscript

37.  Line 186 should read as “Accession numbers of the sequences are listed in Table 1.”

38.  Line 187: Replace “were showed” with “are shown”

39.  Line 190: Replace “are” with “were”. Just curious, is NCBI an abbreviation of the National Library of Medicine?

40.  Line 194: Replace “in” with “at”

41.  Line 197: Replace “in a final concentration” with “at a final concentration”

42.  Line 204: Replace “with a density” with “at a density”

43.  Line 206: Replace “hungered” with “suspended”

44.  Line 207 should read as “The cells were treated with L15 as the control group or with different concentrations of L-glutamate (0, 0.1, 1, 10, and 100 mM) for 1 h as the L-Glu group”

45.  Line 214 should read as “The CCK-8 assay was used to assess cell activity and survival”

46.  Line 226: Replace “using” with “by”

47.  Line 229, 236: Replace “by” with “using”

48.  Line 240: Delete a full-stop at the end of “1.0 x 106”

49.  Line 243: Change “Vector” to “vector”

50.  Line 268-271 should read as “Protein was extracted from the brain cells of mandarin fish, and protein concentrations were determined using a bicinchoninic acid (BCA) protein assay (Biosharp, China)”

51.  Line 289: Replace “that” with “which”

52.  Line 294 should read as “Scholars have classified fish feeding habits as herbivorous, carnivorous, and omnivorous”.

53.  Line 296: Replace “we had” with “ we performed”

54.  Line 297-299 should read as “The results showed that the carnivorous fish had the highest homology. Based on this, we further explored the role of mGluR4 in feeding regulation in extreme carnivorous mandarin fish at the cellular level”.

Comments on the Quality of English Language

I strongly recommends that the authors let some native English speakers check the manuscript.

Author Response

Reviewer 3 Report (New Reviewer)

Comments and Suggestions for Authors

This study investigated the expression and regulation of the metabotropic glutamate receptor 4 (mGluR4), a umami receptor, in Mandarin fish, a model to understand feeding regulation and the role of this specific receptor. The study suggests NPY is regulated by the mGluR4-Ca2+-ERK1/2 Signaling Pathway based on qPCR, western blot, and the use of antagonists/agonists (glutamate) to the receptor.

The manuscript suffers from poor grammer, and the organization of this manuscript is strange. Why are the results (section 2) before the methods (section 3?). The tables are out of order too - Table 1 should be mentioned first, followed by Table 2, Table 3 etc. (79% identity among mammalian (table 4) - why start with Table 4?)

All the statistics should be complete - F statistic, degrees of freedom, and actual p-value should be reported.

What is a mandarin fish brain cell? There are 100s of cells in the brain - neurons (different types - GABA, glutamate), microglia, astrocytes, etc. What cells are these specifically? The authors need to characterize their cell culture with markers (glial or neuronal).

How do the authors know (mGluR4) is responsible. Glutamate will also stimulate other mGLUTs and these could be involved in NPY regulation. Glutamate is not specific to mGLUT4.

Why were the assays conducted at different time points (some 1 hour and some 2 hour)? This is not consistent and makes interpretation of the data challenging.

RNA was not checked for integrity using a BioAnalyzer – this must be done (or at least a gel). There is no mention of the proper controls for qPCR (no reverse transcriptase or no template control). Genomic DNA (Dnase treatment) was not removed, and the qPCR data is thus not reliable.

The statistical section does not include any statistical information about detecting differences among groups nor post-hoc tests.

Other comments:

Grammer mistakes exist, the manuscript requires moderate edits in this regard. There are other examples, but these are only a few.

Ln 37: change "A large number of literatures showed that..." to "literature shows that"

Ln 64: Furthermore, the brain cells of mandarin fish were applied to assess the effects of L-glutamate on Npy. Brain cells are not applied, but rather "used"

Ln 106 "had no significant difference" should be changed to "was not different".

Ln 130 This sentence makes no sense. "We have known that the mGluR4 perceives L-glutamate to regulate Npy while whether Ca2+ involvement in the process has not yet to be determined.

Is Omega spelled wring in line 222 (company?)

Figure 1. Synteny analysis of mGluR4 gene. The synteny analysis performed by searching gene (need date search conducted and reference for database)

Figure 3 caption needs an explanation for the red triangle in the figure.

Table 1 can go into a supplemental section.

What are the vertical bars in Figure 7? "Vertical bars with lowercase letters represented"

Comments on the Quality of English Language

Needs improvement

Round 2

Reviewer 3 Report (New Reviewer)

Comments and Suggestions for Authors

The authors have done a good job answering my queries.

I suggest the following for transparency:

Please add the biological and technical replicates for each assay to the methods sections.

Please consider adding details on the statistical tests. This is important - p<0.05 is generic and lacks information on strength of details.

Please add information about this mixed cell culture - add 2 or 3 sentences on what it entails (add your response to the cell culturing method so readers understand better this is a mixed-cell population derived from brain tissue.

Comments on the Quality of English Language

It is improved, only a few minor grammar mistakes.

Author Response

This manuscript is a resubmission of an earlier submission. The following is a list of the peer review reports and author responses from that submission.

Round 1

Reviewer 1 Report

Comments and Suggestions for Authors

The topic of your manuscript is of high interest and applicable to the aquaculture sector. You still need to address some points:

1- In the abstract: line17: start with capital letter “mandarin fish”

2- Line 82 in the materials section: you missed ”carp” in enumerating species used for multiple a.a. sequence analysis.

3- Line 99: please double check the abbreviation of Phosphate-buffered solution

4- Line 153: please mention the previous study here using the exact reference, if not published yet you can just write “un-published data”.

5- Line 154: do not use the expression “housekeeping gene” instead follow the guidelines for rtPCR writing by using “internal reference gene”.

6- In RT-PCR: If you selected Rpl13a gene as internal reference gene in your study, why you also included β.actin? does it have a main role in your experiment?

7- Western Blot: please explain the method for protein extraction in details.

8- Table 3: you mentioned in the materials part the use of: “We observed the stability of Beta-2-microglobulin (b2m), tyrosine 3-monooxygenase/ tryptophan 5-monooxygenase activation protein, zeta polypeptide (ywhaz), ribosomal  protein L13a (rpl13a), hydroxymethyl-bilane synthase (hmbs), succinate dehydrogenase  complex, subunit A (sdha), and β-actin in the method described in the previous study” and did not mention the main genes used for RT-PCR.. you need to explain that you tested all these genes for their stability and then chose the most stable one (rpl)

9- Line 297: please replace “we have known” with “ revealed or found” more suitable and scientifically sound.

10- discussion: line 397: please replace “scholars” with “scientists or researchers”

11- Line 400: please specify what species of carnivorous fish.

12- Line 410: please mention the previous results on fish with the reference.

13- The manuscript needs extensive English editing.

14- In the conclusion you mentioned a very important part “which offered a novel strategy to promote food  intake by enrichment feed with appropriate amounts of L-glutamate” this needs to be highlighted more in the discussion part…. Your discussion is not clear enough to highlight the main aim of the experiment.

Comments on the Quality of English Language

the manuscript needs extensive English editing